# Using qualitative research methods to understand how surgical procedures and devices are introduced into NHS hospitals: the Lotus study protocol

Daisy Elliott [iD],[1] Natalie S Blencowe [iD],[1,2] Sian Cousins,[1] Jesmond Zahra [iD],[1] Anni Skilton,[1] Johnny Mathews,[3] Sangeetha Paramasivan,[1] Christin Hoffmann [iD],[1] Angus GK McNair [iD],[1,3] Cynthia Ochieng,[1] Hollie Richards,[1] Sina Hossaini,[1] Darren L Scroggie [iD],[1] Barry Main [iD],[1,2] Shelley Potter [iD],[1] Kerry Avery [iD],[1] Jenny Donovan,[4] Jane M Blazeby [iD][1,2]

[1]National Institute for Health Research Bristol Biomedical Research Centre, Surgical Innovation Theme, Centre for Surgical Research, University of Bristol, Bristol, UK
[2]University Hospitals Bristol and Weston NHS Foundation Trust, Bristol, UK
[3]North Bristol NHS Trust, Bristol, UK
[4]Population Health Sciences, Bristol Medical School, University of Bristol, Bristol, UK

**Correspondence to**
Dr Daisy Elliott;
daisy.elliott@bristol.ac.uk

## ABSTRACT

**Introduction** The development of innovative invasive procedures and devices are essential to improving outcomes in healthcare. However, how these are introduced into practice has not been studied in detail. The Lotus study will follow a wide range of 'case studies' of new procedures and/or devices being introduced into NHS trusts to explore what information is communicated to patients, how procedures are modified over time and how outcomes are selected and reported.

**Methods and analysis** This qualitative study will use ethnographic approaches to investigate how new invasive procedures and/or devices are introduced. Consultations in which the innovation is discussed will be audio-recorded to understand information provision practice. To understand if and how procedures evolve, they will be video recorded and non-participant observations will be conducted. Post-operative interviews will be conducted with the innovating team and patients who are eligible for the intervention. Audio-recordings will be audio-recorded, transcribed verbatim and analysed thematically using constant comparison techniques. Video-recordings will be reviewed to deconstruct procedures into key components and document how the procedure evolves. Comparisons will be made between the different data sources.

**Ethics and dissemination** The study protocol has Health Research Authority (HRA) and Health and Care Research Wales approval (Ref 18/SW/0277). Results will be disseminated at appropriate conferences and will be published in peer-reviewed journals. The findings of this study will provide a better understanding of how innovative invasive procedures and/or devices are introduced into practice.

## Strengths and limitations of this study

► Studying innovation in real-time is novel and enables in-depth and contemporaneous insights.
► Including a range of specialities and hospitals will provide a broad overview of how innovative invasive procedures and devices are introduced into NHS hospitals.
► Triangulation of multiple methods of data collection will facilitate a comprehensive understanding of surgical innovation and help to validate findings.
► Findings may be limited by snowball sampling and the self-selecting healthcare professionals who agree to take part in the study.
► Although this study will provide important and rich insights into practices in the UK, findings may not be generalisable to other countries and different healthcare systems.

## INTRODUCTION

At least 12.5 million invasive procedures are delivered in the UK annually.[1] Innovation is critical and constantly occurring because of the drive to improve a range of outcomes. Common targets include using less invasive techniques; shortening hospital stays; improving patient outcomes or creating novel therapeutic options.[2–4] Innovation also frequently occurs because of a healthcare professional's desire to pioneer new techniques and to be recognised as a leader in the field.[4] An independent review by the Royal College of Surgeons of England has predicted that surgery will undergo a massive transformation because of developments in new technologies such as robotics, artificial intelligence, genomics and digital technologies, and concluded that the world of surgery is embarking on a time of innovation.[5]

Unlike the pharmaceutical industry, where there are extensive regulatory and ethical requirements,[6–8] the introduction of innovative invasive procedures and/or devices (IP/Ds) is not adequately regulated. The Idea, Development, Exploration, Assessment and Long-Term Study (IDEAL) framework[9]

outlines five consecutive stages of development, evaluation and dissemination of innovative IP/Ds, although its uptake has been slow[10 11] and it is often incorrectly followed.[12] Furthermore, while innovative IP/Ds should be introduced in the context of formal research studies or in accordance with local hospital policies and processes,[13] ethical approval is rarely gained[8] and procedures can be introduced without any formal governance from local hospitals.[14 15] Important questions have also been raised surrounding informed consent for innovative IP/Ds. For instance, an inquest into the death of the first patient in the UK to undergo robotically assisted heart surgery found the patient had not been fully informed about the comparative risks of robotic versus conventional open surgery.[14]

A recent independent review has highlighted the need for better governance and evaluation of surgical innovation.[16] As little is known about mechanisms of how new IP/Ds are introduced into practice,[17] a greater understanding of how innovative procedures are introduced is imperative to developing guidance and regulation. Complex phenomena such as innovative IP/Ds can be explored in-depth using qualitative methodology so that rich insights and experiences of innovation from a number of perspectives can be captured and explored in detail within their original setting.[18 19] The Lotus study will use qualitative research methods to provide in-depth, real-time insights to how innovative IP/Ds are introduced into National Health Service (NHS) hospitals. Specific objectives of the Lotus study are to explore: (1) how information is communicated to patients and how patients understand this information in order to inform their decision-making process; (2) how procedures evolve over time and (3) how outcomes are selected, measured and reported.

## Methods and analysis

### Design

The Lotus study will use ethnographic approaches to investigate how new IP/Ds are introduced in the NHS. A grounded theory methodology will enable the inductive identification of themes that are derived or grounded in the data.[20 21] Its central principle is of constant comparison, where new findings are systematically compared with existing data so that similarities and differences can be identified through the ongoing assimilation of data.[20 22]

### Case study eligibility

A senior academic surgeon and the Bristol Biomedical Research Centre (BRC) Surgical Innovation theme lead (JMB) will approach individuals from NHS trusts, NHS New Procedure Committees and funding bodies to identify innovative IP/Ds. First, an in-depth qualitative interview with the innovator (a 'background' interview) will be conducted to learn more about the innovative IP/D. Alongside this, the research team will undertake a literature review to identify existing published articles on the procedure. A data extraction form will be developed specifically for the review to record details of each article (relating to authors, year, country of origin, study design, participants and number of participating centres).

Procedures will be eligible if they are deemed to be innovative IP/Ds (see table 1). The background interview and review of published evidence will be examined together by the study team and reasons for exclusion will be documented (ie, if published evidence was such that the outcomes could reasonably be considered to have been systematically evaluated and reported,[23] or that the number of procedures already delivered in the organisation had surpassed that where it would have been viable to collect data about its introduction).

We will aim to follow a range of case studies to capture innovation in different contexts, including varying stages of innovation (as identified by the IDEAL framework), type of innovation (procedure or device), surgical

| Table 1 Case study eligibility | | |
|---|---|---|
| **Procedures will be eligible for the study if they are deemed to be:** | | |
| Innovative | ► | A new or modified procedure that differs from currently accepted local practice, the outcomes of which have not been fully systematically evaluated and reported in a standardised manner, and which may entail unknown risks to the patient.[23 39] This can include the introduction of entirely new procedures as well as undertaking modifications to existing techniques. |
| And: | | |
| An invasive procedure or a device | ► | An invasive procedure will be defined as one in which purposeful and deliberate access to the body is gained via an incision or percutaneous puncture, instrumentation is used in addition to a puncture needle or instrumentation occurs via a natural orifice. It begins when entry to the body is gained and ends when the instrument is removed and/or the skin is closed. It is performed by trained healthcare professionals using instruments, which include, but are not limited to, endoscopes, catheters, scalpels, scissors, devices and tubes.[40] |
| | ► | A device is defined as any instrument, apparatus, appliance, software, material or other article, whether used alone or in combination, which is intended by the manufacturer to be used for human beings.[41] |

specialty and NHS trust type (eg, geographical area, foundation status and acute trust type). Where possible, we will try to identify innovative IP/Ds that are due to be introduced into practice to enable us to capture data from the first time they are performed.

While the exact numbers of participants recruited will vary depending on the frequency of procedures and the number of healthcare professionals involved, we estimate that for each case study we will recruit a minimum of five patients and three healthcare professionals. Although we anticipate approximately 5–10 complete case studies will be included, identification of new case studies will ultimately continue until additional data are not adding anything new to the analytical framework and theoretical saturation is felt to have been achieved. In addition, specific study components may be carried out as standalone elements (eg, video-recordings of operations without the interviews and audio-recordings of consultations or vice versa) for some procedures.

### Recruitment and sampling

For each case study, participants will include healthcare professionals involved in the introduction of an innovative IP/D and patients who are eligible to undergo it. The clinician responsible for its introduction will initially be identified, with additional healthcare professionals being identified by the snowball technique, in which interviewees provide the names of other contacts. Patients will be eligible to take part in the study if they are being offered or have recently undergone (within 3 months from discharge) an innovative IP/D. Patients will be identified by the surgical team and asked to consider taking part in the study. Only patients who are over 18 years old

and have the capacity to consent will be eligible. Patients and healthcare professionals will be asked to provide written consent to taking part in the individual components of the study (eg, audio-recording of consultation, interviews).

## DATA COLLECTION

Each case study will involve the collection of multiple data sources: semi-structured interviews with healthcare professionals and patients, audio-recording consultations between healthcare professionals and patients, video-recording and observing the procedure, and capturing clinical and patient information (these are described below). Data collection and analysis will proceed in parallel, with emerging findings informing data collection.

### Semi-structured interviews

Separate topic guides will be developed for the different types of interviews (see table 2). Discussions will be guided by topic guides to ensure that the discussions cover the same core issues,[24 25] while allowing for probing questions for each participant to enable new issues of importance to be discussed further.[26 27] Topic guides will be adapted as analysis progresses,[24 28] to enable exploration of insights identified across and within case studies. Reflective notes will also be made during the interviews, taking account of the interviewers' thoughts and ideas. All interviews will be audio-recorded using an encrypted recorder.

| Table 2 | Interviews within each case study |
| --- | --- |
| Interview type | Purpose of the interview |
| 'Background' interviews with healthcare professionals | A lead clinician responsible for the introduction of the innovative IP/D will take part in an initial, scene-setting interview to understand what the procedure involves, how it is innovative, any evidence for supporting the use of the procedure, and views as to what patients should be told. Additional healthcare professionals (eg, surgeons, nurses, anaesthetists and representatives involved in regulating the introduction of innovative IP/D at trust or national levels) may also be interviewed to explore any aligning or contrasting experiences and views of the procedure. |
| Post-operative interviews with surgical teams | Surgical teams, including the clinical lead, will be asked to take part in interviews throughout the study to investigate how the IP/D is refined over time, deviations from the planned surgery and plans for modifying the procedure. Where possible, the interview will be conducted immediately after the procedure. |
| Patient interview | Interviews with patients will explore views on the presentation of information provided about the procedure during consultations, reasons underlying decisions to accept or decline the procedure, views/understanding of innovation and (if relevant) their experience of undergoing the procedure and subsequent recovery. |
| End of case study interview | Lead clinicians and other healthcare professionals will be invited to take part in a final interview at the end of the case study period or at the point they, and the study team, consider the procedure to have stabilised (ie, no longer undergoing significant modifications). Interviews will explore their views of, and future plans for, the procedure (ie, further dissemination, training and evaluation). |

IP/D, invasive procedures and/or devices.

### Audio-recording consultations between healthcare professionals and patients

Qualitative analyses of audio-recorded consultations have been successfully applied to the study of informed consent to clinical trial participation[29 30] and will be used in this study to explore how information about innovative IP/Ds is communicated and received. Consultations during which procedures of interest are discussed with eligible patients (including telephone or video-link conversations) will be audio-recorded using an encrypted audio recorder.

### Video-recording and observing the procedure

Consecutive invasive procedures for each case study will be video recorded to investigate the evolution of modification(s) in each procedure over time, and explore how to recognise when a procedure has sufficiently stabilised. Video recording has been done successfully in several other surgical studies[31 32] and has been found to be feasible and acceptable to the surgeon innovators. It is intended that video recordings will capture the entire procedure. Recordings taken from laparoscopic or robotic video feeds will comprise only of unidentifiable intraoperative footage. In the case of 'open' surgery (non-minimally invasive), the field-of-view will capture only the area of interest/surgical site and not identify any healthcare professionals or patients. Audio will not be captured. Should any patient or staff identifiers be inadvertently captured, these will be removed in post-production by the study research photographer. Concurrent observation (where possible) will be non-participant in nature and involve the compilation of field notes relating to verbal and non-verbal communication and contextual factors.

### Clinical and patient data collection

Data about the patients, procedure and nature of the modifications, and clinical outcomes will be recorded from the above data sources and supplemented by hospital records.

## DATA ANALYSIS

In line with the study objectives, a summary of which data sources will be included in the different analyses is in table 3.

### Data analysis of interviews

All audio data will be transcribed verbatim in full or in a targeted manner (where only relevant sections of the consultation will be transcribed). Transcripts will de-identified, checked against original recordings and imported into software (NVivo, QSR International, USA). Interview data will be systematically assigned codes and analysed using constant comparison methods derived from grounded theory methodology.[20 33]

### Data analysis of consultations

We will draw from content, thematic and targeted conversation analytic approaches to investigate the delivery of information during the consultation appointments.[20 34–36] Analysis will centre on exploring patient–healthcare professional interactions (eg, analysis of patient requests for clarification), the type of information communicated to patients (eg, potential risks/benefits, uncertainties), patients' responses to being offered the procedure and their involvement in the decision-making process. Comparisons will be made between the interview data (what surgeons and patients report of the consultation) and the consultation audio data (what transpires in the consultation) across different data sources.

### Data analysis of video recordings and non-participant observations

Each video will be viewed, from beginning to end, by an academic surgeon. This will involve watching and rewatching the recording to familiarise themselves with the procedure and to document movements, instruments, use of any assistants and actions that were captured on the screen.[31] The video will be analysed by the procedural components. This will be done with reference to findings from healthcare professional interviews,

| Table 3 | How the different data sources will be used | | |
|---|---|---|---|
| **Aim:** | **To explore how innovative invasive procedures and/or devices are introduced into NHS hospitals** | | |
| Specific objectives: | To understand how information is communicated to and understood by patients | To understand how procedures evolve over time | To understand how outcomes are selected, measured and reported |
| | ↓ | ↓ | ↓ |
| Data sources included in analysis: | 'Background' interviews<br>Audio-recording consultations<br>Patient interviews | 'Background' interviews<br>Video-recording procedures<br>Non-participant observations<br>Post-operative interviews<br>Clinical and patient information<br>End of case study interview | 'Background' interviews<br>Video-recording procedures<br>Non-participant observations<br>Post-operative interviews<br>Clinical and patient information<br>End of case study interview |

NHS, National Health Service.

alongside an understanding of the existing publications and surgical knowledge of the procedure. Essentially, the research team will 'deconstruct' the intervention into its component parts.[31 37 38] Where available, notes from non-participant observations will be added to the respective operative step(s) from the video recording. Other intra-operative data may be viewed in conjunction during video analysis, such as length of procedure, port-placement time, time on operative console and blood loss. In the case of innovative robotic procedures, images of port-site placements will also be captured and reviewed. This will enable the research team to document how the component parts were delivered, as well as components which were not delivered or modified compared with what was anticipated. It will also identify how unexpected events are managed intraoperatively, and how patient (anatomical) or contextual (theatre) factors are dealt with. Videos will be analysed in chronological order to understand how the procedure evolves over time. Taken together, this will allow the research team to investigate how the procedure is refined and changed. Where changes to the procedure are made, post-operative interviews with the surgeons will be conducted to explore the rationale for the modification. This will compile a descriptive analysis of the novel procedure as a case study.

## Patient and public involvement

The current study comprises a core component of the work undertaken within the National Institute for Health Research BRC Surgical Innovation theme, which aims to improve the safe and transparent translation of innovative IP/Ds to clinical practice. A patient and public involvement (PPI) group will be established, in which patients who have undergone surgery are asked about their views regarding how new surgical procedures are undertaken in NHS clinical practice. The PPI group will have involvement in the Lotus study from inception to its completion. This will include providing feedback on study aims, data collection plans and study documentation as well as contributing to the data analysis and dissemination of findings.

## Ethics and dissemination

The study has been reviewed and approved (HRA and Health and Care Research Wales) by the Frenchay Research Ethics Committee (Ref 18/SW/0277) on 31 December 2018. The initial study length will be 5 years. Study results will be presented to different audiences including academic, clinical and lay members of the public. It is hoped that this work will facilitate a better understanding of the process of surgical innovation, so that new methods for efficient, safe and timely design and conduct of innovative invasive IP/Ds can be developed. Specifically, it is hoped that this will be achieved by: (1) optimising patient information provision and informed consent for innovative IP/Ds; (2) improving the evaluation of surgical interventions and being able to establish studies and study designs that precede and include surgical RCTs; (3) establishing guidance for the selection and reporting of outcomes of innovative IP/Ds and (4) developing a reporting and sharing platform for innovative procedures (an 'e-platform').

**Contributors** DE, NSB and JB developed the idea for the current study. DE drafted the first version of the protocol, with critical revision from JB. NSB, SC, JZ, AS, JM, SP, CH, AGM, CO, HR, SH, DLS, BM, SP, KA and JD commented on versions of the protocol and gave approval for the manuscript to be submitted. The NIHR Bristol Biomedical Research Centre Surgical Innovation theme is directed by JB (theme lead).

**Funding** This study was supported by the NIHR Biomedical Research Centre at University Hospitals Bristol and Weston NHS Foundation Trust and the University of Bristol (BRC-1215-20011). The views expressed in this publication are those of the author(s) and not necessarily those of the NHS, the National Institute for Health Research or the Department of Health and Social Care. JLD and JMB are NIHR Senior Investigators. NSB and AGKM are MRC Clinician Scientists.

**Competing interests** None declared.

**Patient and public involvement** Patients and/or the public were involved in the design, or conduct, or reporting, or dissemination plans of this research. Refer to the Methods section for further details.

**Patient consent for publication** Not applicable.

**Provenance and peer review** Not commissioned; externally peer reviewed.

**ORCID iDs**
Daisy Elliott http://orcid.org/0000-0001-8143-9549
Natalie S Blencowe http://orcid.org/0000-0002-6111-2175
Jesmond Zahra http://orcid.org/0000-0002-7947-2216
Christin Hoffmann http://orcid.org/0000-0002-6293-3813
Angus GK McNair http://orcid.org/0000-0002-2601-9258
Darren L Scroggie http://orcid.org/0000-0002-5472-2602
Barry Main http://orcid.org/0000-0003-0622-805X
Shelley Potter http://orcid.org/0000-0002-6977-312X
Kerry Avery http://orcid.org/0000-0001-5477-2418
Jane M Blazeby http://orcid.org/0000-0002-3354-3330

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
