## [Reviewer comments · BMJ Open]

ARTICLE DETAILS

TITLE (PROVISIONAL)	Using qualitative research methods to understand how surgical procedures and devices are introduced into NHS hospitals: The Lotus study protocol
AUTHORS	Elliott, Daisy; Blencowe, Natalie; Cousins, Sian; Zahra, Jesmond; Skilton, Anni; Mathews, Johnny; Paramasivan, Sangeetha; Hoffmann, Christin; McNair, Angus; Ochieng, Cynthia; Richards, Hollie; Hossaini, Sina; Scroggie, Darren; Main, Barry; Potter, Shelley; Avery, Kerry; Donovan, Jenny; Blazeby, Jane

VERSION 1 – REVIEW

REVIEWER	Marcus, Hani Imperial College London
REVIEW RETURNED	01-Mar-2021

GENERAL COMMENTS	The authors should be commended for an excellent protocol. As with most of the best ideas, it is deceptively simple. It is rare that I have had the opportunity to review a protocol where the topic is as interesting, the text as well structured, and the writing as clear and succinct. My have only two minor comments. First, I would argue for a little more on the recruitment plan - both how you will identify innovations, and how many you will include. While I accept that a power analysis is not appropriate for qualitative studies, there needs to be something about how many innovations you plan to recruit, and when you will close recruitment. Second, I think there needs to be more on the video recordings and their analysis. The authors will recognise this is a huge amount of work. I would suggest more detail on what exactly is being recorded (the operating room, or e.g., a video feed), and the scope of analysis (even surgeons within a department often don't agree on the steps of an established operation).
--

REVIEWER	Wajed, S Royal Devon and Exeter Hospital, Upper GI Surgery
REVIEW RETURNED	15-Mar-2021

GENERAL COMMENTS	The reviewer provided a marked copy with additional comments. Please contact the publisher for full details.
--

REVIEWER	Slack , Mark CMR Surgical Ltd
REVIEW RETURNED	05-Apr-2021

GENERAL COMMENTS	This is an extremely important subject and needs attention and as such deserves publication by a group with good experience in this area. I am not convinced that you will achieve the aims around the video footage and would like clarity to understand how they will anonymize the videos.
---

VERSION 1 – AUTHOR RESPONSE

Reviewer: 1
 Dr. Hani Marcus, Imperial College London Comments to the Author:

Response: Thank you to this reviewer for such positive and encouraging feedback.

My have only two minor comments. First, I would argue for a little more on the recruitment plan - both how you will identify innovations, and how many you will include. While I accept that a power analysis is not appropriate for qualitative studies, there needs to be something about how many innovations you plan to recruit, and when you will close recruitment.

Response: Thank you for raising this point. We have now added more information about how we will identify innovations:

Procedures will be eligible if they are deemed to be innovative IP/Ds (see Table 1). The background interview and review of published evidence will be examined together by the study team and reasons for exclusion will be documented (i.e. if published evidence was such that the outcomes could reasonably be considered to have been systematically evaluated and reported, or that the number of procedures already delivered in the organisation had surpassed that where it would have been viable to collect data about its introduction). (Page 5)

We have also created a table summarising our selection criteria (page 5/6):

Table 1: Case study eligibility

Procedures will be eligible for the study if they are deemed to be:	
Innovative	 This will be defined as a new or modified procedure that differs from currently accepted local practice, the outcomes of which have not been fully systematically evaluated and reported in a standardised manner, and which may entail unknown risks to the patient^{1 2}. This can include the introduction of entirely new procedures as well as undertaking modifications to existing techniques.
And:	
An invasive procedure or a device	 An invasive procedure will be defined as one in which purposeful and deliberate access to the body is gained via an incision or percutaneous puncture, instrumentation is used in addition to a puncture needle, or instrumentation occurs via a natural orifice. It begins when entry to the body is gained and ends when the instrument is removed and/or the skin is closed. It is performed by trained healthcare professionals using instruments, which include, but are not limited to, endoscopes, catheters, scalpels, scissors, devices and tubes³.

	 • A device is defined as any instrument, apparatus, appliance, software, material or other article, whether used alone or in combination, which is intended by the manufacturer to be used for human beings⁴.
--	--

We have added more information about our anticipated sample sizes:

Whilst the exact numbers of participants recruited will vary depending on the frequency of procedures and the number of healthcare professionals involved, we estimate that for each case study we will recruit a minimum of five patients and three healthcare professionals. Although we anticipate approximately 5-10 complete case studies will be included, identification of new case studies will ultimately continue until additional data are not adding anything new to the analytical framework and theoretical saturation is felt to have been achieved. In addition, specific study components may be carried out as stand-alone elements (e.g. video-recordings of operations without the interviews and audio-recordings of consultations, or vice versa) for some procedures. (Page 6)

We have also added a statement about the duration of the study:

The study has been reviewed and approved (HRA and Health and Care Research Wales) by the Frenchay Research Ethics Committee (Ref 18/SW/0277) on 31/12/2018. The initial study length will be five years. (Page 9)

Second, I think there needs to be more on the video recordings and their analysis. The authors will recognise this is a huge amount of work. I would suggest more detail on what exactly is being recorded (the operating room, or e.g., a video feed), and the scope of analysis (even surgeons within a department often don't agree on the steps of an established operation).

Response: We have expanded the section in the protocol to explain what exactly is being recorded: It is intended that video recordings will capture the entire procedure. Recordings taken from laparoscopic or robotic video feeds will comprise only of unidentifiable intra-operative footage. In the case of 'open' surgery (non-minimally invasive), the field-of-view will capture only the area of interest/surgical site and not identify any healthcare professionals or patients. Audio will not be captured. Should any patient or staff identifiers be inadvertently captured, these will be removed in post-production by the study research photographer. Concurrent observation (where possible) will be non-participant in nature and involve the compilation of field notes relating to verbal and non-verbal communication and contextual factors. (Page 8)

We have also provided more information about the plans for video analysis, and hope this is clearer now: Video recording has been done successfully in several other surgical studies 34 35 and has been found to be feasible and acceptable to the surgeon innovators. (Page 8)

Each video will be viewed, from beginning to end, by an academic surgeon. This will involve watching and re-watching the recording to familiarise themselves with the procedure and to document movements, instruments, use of any assistants and actions that were captured on the screen³⁴. The video will be analysed by the procedural components. This will be done with reference to findings from healthcare professional interviews, alongside an understanding of the existing publications and surgical knowledge of the procedure. Essentially, the research team will 'deconstruct' the intervention into its component parts^{34 40 41}. Where available, notes from non-participant observations will be added to the respective operative step(s) from the video recording. Other intra-operative data may be viewed in conjunction during video analysis, such as length of procedure, port-placement time, time on operative console and blood loss. In the case of innovative robotic procedures, images of port-site placements will

also be captured and reviewed. This will enable the research team to document how the component parts were delivered, as well as components which those were not delivered or modified compared to what was anticipated. It will also identify how unexpected events are managed intra-operatively, and how many patient (anatomical) or contextual (theatre) factors are dealt with. Videos will be analysed in chronological order to understand how the procedure evolves over time. Taken together, this will allow the research team to investigate how the procedure is refined and changed. Where changes to the procedure are made, post-operative interviews with the surgeons will be conducted to explore the rationale for the modification. This will compile a descriptive analysis of the novel procedure as a case study. (Page 9/10)

Reviewer: 2

Dr. S Wajed, Royal Devon and Exeter Hospital, University of Exeter Medical School

Response: Thank you for your thoughtful comments. We agree it is a complex area. This exploratory study is the first step to understanding how innovative procedures are introduced and change in practice. It compliments other work we are undertaking looking at current practice for governance³ and informed consent⁵ for new procedures. Our aim is to understand and develop methods – so drawing conclusions at this stage is not expected.

Outside the NHS, medical advances can be introduced providing the clinician and healthcare organisation are satisfied with regards to safety and clinical efficacy, and that there is sufficient demand to make its provision financially and commercially viable. Within the NHS however, financial restrictions mean that it is incumbent on a clinician or specialist to make an argument for the introduction a new intervention. A business model is normally required, and an NHS Management team consider whether any additional financial burden or resource allocation is justifiable. This study would therefore be enhanced with involvement of NHS Management team members interviews in addition to those of clinicians and patients.

Response: Thank you for this helpful suggestion. In a previous study⁵, we have conducted interviews with those involved in regulating the introduction of new/modified procedures and/or devices at trust or national levels to understand what processes were in place for the introduction of new procedures. We found that participants found it challenging to define surgical innovation there was variation in how NHS trusts govern the introduction of new procedures.

We agree that interviewing this group this would provide important insights in the current study. We have now updated the manuscript so that it states that individuals involved in the decision-making process, such as NHS Management team members, may also be interviewed:

Additional healthcare professionals (e.g. surgeons, nurses, anaesthetists and representatives involved in regulating the introduction of innovative IP/D at trust or national levels) may also be interviewed to explore any aligning or contrasting experiences and views of the procedure. (Page 9)

The Authors highlight the wide spectrum of innovations that are constantly under development applicable across all healthcare specialties. The protocol does not specify how many different interventions will be studied, nor from which specialties they might be chosen from. It would be important to try and represent several diverse medical specialties, as well as explore different areas of innovation.

Response: We have now added more information about how many innovations we estimate we will study:

Whilst the exact numbers of participants recruited will vary depending on the frequency of procedures and the number of healthcare professionals involved, we estimate that for each case study we will recruit a minimum of five patients and three healthcare professionals. Although we anticipate approximately 5-10 complete case studies will be included, identification of new case studies will ultimately continue until additional data are not adding anything new to the analytical framework and theoretical saturation is felt to

have been achieved. In addition, specific study components may be carried out as stand-alone elements (e.g. video-recordings of operations without the interviews and audio-recordings of consultations, or vice versa) for some procedures. (Page 6)

We have stated that we hope to capture different types/phases of innovations from a range of specialties:

We will aim to follow a range of case studies to capture innovation in different contexts, including varying stages of innovation (as identified by the IDEAL framework⁹), type of innovation (procedure or device), surgical specialty, and NHS trust type (geographical area, foundation status and acute trust type).(Page 6)

It does not clearly explain what kind of interventions will be chosen or what the selection criteria might be. Some form of rationale for choosing which examples to study should therefore be offered.

Response: Thank you for pointing this out – we agree that we should have been clearer about this. We have now added more information about the selection process and also included a table with the specific selection criteria:

Procedures will be eligible if they are deemed to be innovative IP/Ds (see Table 1). The background interview and review of published evidence will be examined together by the study team and reasons for exclusion will be documented (i.e. if published evidence was such that the outcomes could reasonably be considered to have been systematically evaluated and reported, or that the number of procedures already delivered in the organisation had surpassed that where it would have been viable to collect data about its introduction). (Page 5)

Table 1: Case study eligibility

Procedures will be eligible for the study if they are deemed to be:	
Innovative	 This will be defined as a new or modified procedure that differs from currently accepted local practice, the outcomes of which have not been fully systematically evaluated and reported in a standardised manner, and which may entail unknown risks to the patient^{1 2}. This can include the introduction of entirely new procedures as well as undertaking modifications to existing techniques.
And:	
An invasive procedure or a device	 An invasive procedure will be defined as one in which purposeful and deliberate access to the body is gained via an incision or percutaneous puncture, instrumentation is used in addition to a puncture needle, or instrumentation occurs via a natural orifice. It begins when entry to the body is gained and ends when the instrument is removed and/or the skin is closed. It is performed by trained healthcare professionals using instruments, which include, but are not limited to, endoscopes, catheters, scalpels, scissors, devices and tubes³. A device is defined as any instrument, apparatus, appliance, software, material or other article, whether used alone or in combination, which is intended by the manufacturer to be used for human beings⁴.

Once appropriate selections have been made, there should then be consideration as to which communities the studies are to be undertaken in. This is of relevance given the ethnographic approach proposed. There is considerable variation across the NHS of the population base being served and the resources within each individual hospital. These factors may well influence the ability for innovation to be introduced and established. These institutions should be representative of the inherent regional, socio-economic, and financial diversity that occurs in across NHS provision. Ideally each intervention selected should also be examined in several different ethnographic settings so that comparisons can be made. The Authors should indicate how many different interventions they intend to examine, and in how many settings each will take place in.

Response: Thank you for raising this important point – we agree that we should capture innovations across a range of NHS settings. We have now included this in our sampling strategy: We will aim to follow a range of case studies to capture innovation in different contexts, including varying stages of innovation (as identified by the IDEAL framework⁹), type of innovation (procedure or device), surgical specialty, and NHS trust type (e.g. geographical area, foundation status and acute trust type). (Page 6)

Due to the exploratory nature of the study, we are not able to determine how many settings each case study will take place in. As described above, we have stated our sampling strategy will be purposive and will aim to seek out a range of case studies to capture innovation in different contexts. We have also added more information about our anticipated sample size:

Although we anticipate approximately 5-10 complete case studies will be included, identification of new case studies will ultimately continue until additional data are not adding anything new to the analytical framework and theoretical saturation is felt to have been achieved. (Page 6)

There ought to be a formal and open process by which, once a particular intervention has been selection for case study, that clinicians from all NHS organisations are invited to apply to take part and share experiences. There may be a potential risk of bias if the current proposal of Authors selecting clinicians, and then those individuals recommending others (“snowballing”) were to be undertaken. Requesting interest to join in the Study could easily be facilitated via the relevant Specialty organisations.

Response: We have now added a sentence to the study summary box, acknowledging the limitations of our sampling strategy:

- Findings may be limited by snowball sampling and the self-selecting healthcare professionals who agree to take part in the study

Satisfactory outcomes, including clinical benefit, adverse events and financial considerations depend on the type of intervention examined. In some situations, clinical outcomes will not become apparent until several months or years. The Authors should therefore give some form of indication as to the overall timeframe that the study will take place in.

Response: Although, Lotus will study the process of new procedures being introduced (rather than evaluating the clinical outcomes of the procedures), we agree it is important to provide an indication of the overall timeframe. We have now clarified the duration of the study:

The study has been reviewed and approved (HRA and Health and Care Research Wales) by the Frenchay Research Ethics Committee (Ref 18/SW/0277) on 31/12/2018. The initial study length will be five years. (Page 9)

Overall, the question of how new medical technology and advances are introduced into the NHS is important to try and understand. Successful completion of this study could help advance our knowledge in this area. Restrictions which are currently limiting the provision of potentially beneficial innovative treatment options in the NHS may be denying access to healthcare provision to individuals, based on financial status, geographical location, status of their healthcare provider and the enthusiasm of their nominated clinician.

I would like to thank the Editors of BMJ Open for inviting me to undertake this review, and hope that the Authors will find these comments constructive.

Response: Thank you again for such positive and constructive feedback.

Reviewer: 3

Dr. Mark Slack , CMR Surgical Ltd, Addenbrooke's Hospital

Response: Thank you for such encouraging feedback.

I am not convinced that you will achieve the aims around the video footage and would like clarity to understand how they will anonymize the videos.

Response: We agree this is an ambitious project, and have added a statement to highlight the exploratory nature of this work:

Video recording has been done successfully in several other surgical studies 34 35, and it has been found to be feasible and acceptable to the surgeon innovators. (Page 8)

We have also explained how we will anonymise the video-recordings, and hope this is clearer now: Recordings taken from laparoscopic or robotic video feeds will comprise only of unidentifiable intra-operative footage. In the case of 'open' surgery (non-minimally invasive), the field-of-view will capture only the area of interest/surgical site and not identify any healthcare professionals or patients. Audio will not be captured. Should any patient or staff identifiers be inadvertently captured, these will be removed in post-production by the study research photographer. (Page 8)

We have provided more information about we plan to meet our objectives using data from the video-recordings:

Each video will be viewed, from beginning to end, by an academic surgeon. This will involve watching and re-watching the recording to familiarise themselves with the procedure and to document movements, instruments, use of any assistants and actions that were captured on the screen³⁴. The video will be analysed by the procedural components. This will be done with reference to findings from healthcare professional interviews, alongside an understanding of the existing publications and surgical knowledge of the procedure. Essentially, the research team will 'deconstruct' the intervention into its component parts^{34 40 41}. Where available, notes from non-participant observations will be added to the respective operative step(s) from the video recording. Other intra-operative data may be viewed in conjunction during video analysis, such as length of procedure, port-placement time, time on operative console and blood loss. In the case of innovative robotic procedures, images of port-site placements will also be

captured and reviewed. This will enable the research team to document how the component parts were delivered, as well as components which those were not delivered or modified compared to what was anticipated. It will also identify how unexpected events are managed intra-operatively, and how many patient (anatomical) or contextual (theatre) factors are dealt with. Videos will be analysed in chronological order to understand how the procedure evolves over time. Taken together, this will allow the research team to investigate how the procedure is refined and changed. Where changes to the procedure are made, post-operative interviews with the surgeons will be conducted to explore the rationale for the modification. This will compile a descriptive analysis of the novel procedure as a case study. (Page 9/10)

References

1. Biffi WL, Spain DA, Reitsma AM, et al. Responsible Development and Application of Surgical Innovations: A Position Statement of the Society of University Surgeons. *J Am Coll Surg* 2008;206(6):1204-09. doi: <https://doi.org/10.1016/j.jamcollsurg.2008.02.011>
2. Birchley G, Huxtable R, Ives J, et al. Have We Made Progress in Identifying (Surgical) Innovation? *The American Journal of Bioethics* 2019;19(6):25-27. doi: 10.1080/15265161.2019.1602181
3. Cousins S, Blencowe NS, Blazeby JMJB. What is an invasive procedure? A definition to inform study design, evidence synthesis and research tracking. 2019;9(7):e028576.
4. Medical Devices Regulations 2002 (SI 2002 No 618 aaUM. [Available from: <https://www.gov.uk/guidance/medical-devices-how-to-comply-with-the-legal-requirements> accessed 11th January 2021. .
5. Zahra J, Paramasivan S, Blencowe NS, et al. Discussing surgical innovation with patients: a qualitative study of surgeons' and governance representatives' views. 2020;10(11):e035251.

VERSION 2 – REVIEW

REVIEWER	Wajed, S Royal Devon and Exeter Hospital, Upper GI Surgery
REVIEW RETURNED	14-Jul-2021
GENERAL COMMENTS	Interesting novel qualitative approach in attempting to study a complex area